# The Influence of Copper Nanoparticles on Neurometabolism Marker Levels in the Brain and Intestine in a Rat Model

**DOI:** 10.3390/ijms241411321

**Published:** 2023-07-11

**Authors:** Monika Cendrowska-Pinkosz, Magdalena Krauze, Jerzy Juśkiewicz, Bartosz Fotschki, Katarzyna Ognik

**Affiliations:** 1Chair and Department of Human Anatomy, Medical University of Lublin, 20-090 Lublin, Poland; 2CM Alergologia, 20-865 Lublin, Poland; 3Department of Biochemistry and Toxicology, Faculty of Animal Science and Bioeconomy, University of Life Sciences in Lublin, 20-950 Lublin, Poland; 4Department of Biological Functions of Food, Division of Food Science, Institute of Animal Reproduction and Food Research of the Polish Academy of Sciences, 10-748 Olsztyn, Poland

**Keywords:** copper nanoparticles, copper carbonate, brain, jejunum, neurodegeneration

## Abstract

The aim of this study is to assess the effect of different forms and dosages of copper on the levels of markers depicting the neurodegenerative changes in the brain and the jejunum. The experiment was performed using 40 male Wistar rats fed a typical rat diet with two dosages of Cu used as CuCO_3_ (6.5 and 13 mg/kg diet) and dietary addition of two CuNP dosages (standard 6.5 and enhanced 13 mg/kg diet), randomly divided into four groups. The levels of neurodegenerative markers were evaluated. Nanoparticles caused a reduction in the level of glycosylated acetylcholinesterase (GAChE), an increase the level of acetylcholinesterase (AChE) and lipoprotein receptor-related protein 1 (LRP1), a reduction in β-amyloid (βAP) in the brain and in the intestine of rats and a reduction in Tau protein in the brain of rats. The highest levels of AChE, the ATP-binding cassette transporters (ABC) and LRP1 and lower levels of toxic GAChE, β-amyloid, Tau, hyper-phosphorylated Tau protein (p-Tau) and the complex of calmodulin and Ca^2+^ (CAMK2a) were recorded in the tissues of rats receiving a standard dose of Cu. The neuroprotective effect of Cu can be increased by replacing the carbonate form with nanoparticles and there is no need to increase the dose of copper.

## 1. Introduction

Copper is essential in modulating neuronal proliferation and is involved in many important processes in the body. An example is the regulation of biochemical reactions, such as cellular oxidation or free radicals’ protection. In neurons, the accumulation of free radicals destabilizes both the cellular membranes and the function of mitochondria, which results in energetic disturbances and changes in neurotransmission [1,2,3,4]. Nervous tissue metabolism is very intensive and relates to the need to produce a large amount of ATP during oxidative phosphorylation. The effect of such a high rate of reactions in the respiratory chain intensification is the generation of large amounts of free radicals and oxidative stress to which the nervous tissue is particularly sensitive (greater demand for ATP in these cells compared to others and limiting the possibility of internal DNA repair). Neurometabolism disruption is particularly dangerous because nervous system cells have a very limited capacity for glycolysis and do not have a reserve energy supply system; therefore, cellular respiration can be disturbed [5,6]. Because of neurometabolism changes, neurons die [4,5,6,7]. In diet, copper is present in an oxidized form of Cu^2+^ [8] and the chemical form of copper plays a crucial role in its utility. Copper is highly reactive in biological systems because it can be both a donor and an acceptor of electrons [3,8]. In the serum and inside the cells, copper exists in the free form (Cu^2+^), and it is believed that this form is absorbed into the cell by the divalent metal transporter [4]. Peña et al. [9] report that the chemical form and dosage of copper in the diet strongly affect its influence on processes that occur in the body. The study of Cu^2+^ speciation in extracellular fluids is paramount to understand Cu metabolism, homeostasis, and the pathophysiology of Cu-related diseases [10]. Current reports suggest that both an excess and deficiency of copper can be harmful; therefore, it is important to maintain the homeostasis of this metal [3,4,6,8]. An impaired balance of copper in the body can accelerate the aging process, promote neurodegeneration and tumorigenesis, and perpetuate unfavorable genetic mutations [3,8]. A deficiency of this element impaired cellular respiration and showed a dysregulation of energy metabolism, and dysfunction of the immune system is also observed. An excess of copper can support the overproduction of free radicals, thus impairing cellular respiration and neurometabolism. Then, a cascade of reactions is triggered, associated with the increased production of toxic neurodegenerative structures, i.e., β-amyloid, hyperphosphorylated Tau protein and glycosylated acetylcholinesterase, epigenetic changes are stimulated, and DNA repair mechanisms are impaired [11,12]. According to Patel et al. [13], excess inorganic copper can accumulate in the brain, especially in the hippocampus, which is a crucial spot of neurogenesis. Bagheri et al. [8] state that the hippocampus is the first and most often damaged part of the brain and its dysfunction and degeneration caused by excessive amounts of copper lead to the rapid progression of neuronal degeneration. A more thorough understanding of the pathomechanisms of these changes is aimed at improving the diagnostic process, as well as finding new potential therapeutic targets. Determining the optimal dose of copper to protect neurometabolism is very difficult due to the presence of the blood–brain barrier and the difficult penetration of copper into the brain. On the one hand, copper has a beneficial and necessary effect, because being an element of the path that protects the cell against the negative effects of oxidative stress in cells, it reduces the generation of free radicals that destabilize cell membranes and participates in neurometabolism [6,7].

This issue is particularly visible in the endothelial cells that line the blood vessels of the brain. Due to the high reactivity of inorganic copper (carbonates, chlorides, sulfates), it is important to search for different sources and forms of copper that are easily absorbed in the digestive system. Copper reactivity allows the cations to easily interact with different elements of the diet, mostly through initiating oxidation [14,15,16,17]. The binding of copper ions by plasma proteins often makes them poorly permeable through the blood–brain barrier. The combination with nanoparticles may be one of the therapeutic options, thanks to which small, therapeutic amounts of nanocopper could more easily penetrate the blood–brain barrier and reach the brain. There are obstacles regarding the restriction of neurodegenerative changes and the treatment of neurological disorders due to a weak penetration of copper into the brain through the brain–blood barrier. Strategies supporting the transport of copper across the blood–brain barrier are therefore needed to effectively combat these conditions.

The results of experiments on the use of nanoparticles look very promising. Nanocopper enclosed in nanoparticles and administered peripherally has a chance to penetrate the blood–brain barrier much more easily than the ionic form [12,14,18,19]. This knowledge opens an important new area for potential therapeutic interventions based on appropriate supplementation of copper in the prevention and treatment of neurodegenerative diseases.

The main neurodegenerative changes include neuronal death caused by the accumulation of neurodegeneration structures. These include senile plaques made from β-amyloid protein (βAP) and well as neurofibrillary tangles of hyper-phosphorylated Tau protein (p-Tau). Increasing the concentration of pathological proteins, mainly β-amyloid, Tau and p-Tau and their toxic derivative, glycosylated acetylcholinesterase (GAChE), in the cytoplasm results in the damage and death of neurons. The amyloid plaques and neurofibrillary tangles of Tau protein form in between the neurons and block the communication between them, disrupting their metabolism and repair processes. They impede a neuron’s ability to effectively conduct nervous impulses, which leads to neurodegeneration and neuronal death [19,20]. The key enzyme ensuring the proper flow of nerve impulses in synapses and protecting neurons is acetylcholinesterase (AChE). When AChE binds with the neurotoxic βAP and Tau protein, they form another harmful structure called glycosylated acetylcholinesterase (GAChE). In terms of AChE, both its excess and deficiency have a neurodegenerative effect. The fluctuations in AChE levels and the appearance of GAChE can promote mitochondrial dysfunction and give rise to high amounts of reactive oxygen species (ROS) which induce apoptotic neuronal death [21]. Because of the lipid structure of the neural tissue, the ATP-binding cassette transporters (ABC) are essential in its functioning. The presence of ABC transporters allows for the transport of lipids in the brain and, therefore, maintains homeostasis. Lipoprotein receptor-related protein 1 (LRP1) is another important factor that helps degrade the toxic βAP and protect neurons [19,20]. Neurometabolism is regulated by the complex of calmodulin and Ca^2+^ (CAMK2a). This enzyme may excessively adversely activate protein kinases responsible for the phosphorylation in synaptic mitochondria and the appearance of p-Tau. To measure the severity of neurodegeneration caused by epigenetic changes, it might be useful to verify the level of methylation in the DNA [22,23,24].

The presented research provides new knowledge that fits into the current trend of research on the relationship between the level and chemical form of supplemented copper and the development of neurodegenerative disorders. There is still a lack of information and full conclusions on how copper in the form of Cu^2+^ may imply pathological processes in the nervous tissue, and many of them remain poorly understood.

In this study, we proposed that the neuroprotective properties of dietary copper can be enhanced when the traditionally administered carbonate form is exchanged for copper nanoparticles (CuNPs) and their influence can be dependent on the applied dosage. The aim of this study was to assess the effect of different forms and dosages of copper applied to the rats’ diet on the levels of markers depicting the neurodegenerative changes in the brain and the jejunum.

## 2. Results

Neurometabolism changes and the accumulation of toxic peptides and protein complexes may result in deterioration of rats’ health. The results of the conducted tests do not indicate a negative effect of the dose or form of copper on the health condition of rats. There was also no decrease in the final body weight of rats from the groups receiving nanocopper compared to the control group. The animals did not show any negative behavioral features associated with receiving copper nanoparticles, for both the 6.5 mg and 13 mg doses.

Results of the two-way ANOVA analysis showed that acetylcholinesterase (AChE) showed a Source × Dose (S × D; *p* = 0.012) interaction in the rat brain. It is demonstrated by the fact that, regardless of the source of copper, applying a higher dosage of CuNPs has increased the level of AChE in the CuNPs_13_ group, whereas in the CuSALT_13_ group, the levels of AChE decreased (Table 1). AChE is a key enzyme for the proper conduction of nerve impulses and the protection of neurons. In the case of a decrease in the level of acetylcholinesterase, or an excessive increase in the level of this neuroregulator, a clear disturbance of the nervous tissue is observed.

### 2.1. The Effects of Different Sources of Copper

Administering two different sources of copper (CuCO_3_ and CuNPs) influenced the variability of neurometabolic markers in the rat’s brain and jejunum. Notably, CuNPs had a stronger influence on decreasing the levels of GAChE both in the brain (*p* = 0.009) and the jejunum (*p* = 0.007) (Table 1 and Table 2). Glycosylated acetylcholinesterase (GAChE) is an inactive complex of the acetylcholinesterase enzyme with β-amyloid and hyperforylated Tau protein (p-Tau). It is formed under the influence of the complex of calmodulin and Ca^2+^ (CAMK2a) when there are changes in neurometabolism. The most common cause is amyloid oligomerization and abnormal phosphorylation of the Tau protein in the mitochondrion.

The use of CuNPs resulted in a reduction in the level of β-amyloid protein in the brain (*p =* 0.017) and in the jejunum of rats (*p =* 0.05). Toxic amyloid-β is formed as a result of amyloid oligomerization (Table 3 and Table 4). Clusters of this peptide form senile plaques that destabilize neurometabolism. Applying CuNPs regardless of the applied dosage caused a decrease in Tau protein levels in the brain (*p =* 0.001). Excessive accumulation of Tau protein in neurons is the initiator of neurodegenerative changes. It is the cause of degeneration in the earliest stages of the disease. β-amyloid and p-Tau appear in successive stages. Hyperphosphorylated Tau protein impairs the functioning of neuronal mitochondria. Furthermore, it was observed that administering CuNPs increased the levels of AChE (*p* = 0.024) and LRP1 (*p =* 0.043) in the rat’s jejunum (Table 2). Low-density lipoprotein receptor-related protein 1 (LRP1) supports the degradation of toxic amyloid-β. There was no adverse effect of the addition of copper nanocasts on the level of p-Tau, ABC, CAMK2a and the percentage of DNA methylation (Table 1, Table 2, Table 3 and Table 4).

### 2.2. The Effects of Different Dosages of Copper

Administering two different doses of copper in the diet (6.5 mg and 13 mg) significantly influenced the levels of neurometabolic markers in the rat’s brain and jejunum. There was a statistically significant increase in AChE levels after administering 6.5 mg of copper (Table 1 and Table 2). The levels of GAChE and βAP decreased both in the brain (*p* = 0.026; *p* = 0.036) and the jejunum (*p* = 0.021; *p* = 0.007) after administering 6.5 mg of copper (Table 1, Table 2, Table 3 and Table 4). Regardless of the chemical form, administering 6.5 mg of copper resulted in notably lower concentrations of p-Tau (*p* = 0.003) and CAMK2a (*p* = 0.005) in the brain (Table 1). What is more, administering a lower dosage of copper resulted in an increase in ABC level in the brain (*p* = 0.033) and an increase in LRP1 in the jejunum (*p* = 0.002) (Table 1 and Table 2). ATP-binding cassette transporters enable the transport of lipids in the brain and determine homeostasis. They also participate in the process of ATP hydrolysis in the cell and are responsible for the translocation of nutrients through the membrane.

The level of Tau protein was significantly lower in the jejunum of rats who received 6.5 mg of copper compared with animals who received 13 mg of copper in their diet. There were statistically significant increases in AChE levels after the administration of 6.5 mg of copper in the brain and intestine (Table 1, Table 2, Table 3 and Table 4). In the brain, the level of ABC was found to increase after the administration of nanocopper at a dose of 6.5 mg (*p* = 0.033). This relationship was not found in the jejunum (*p* = 0.001).

## 3. Discussion

Because excess copper in the diet may have neurotoxic effects, it is worth assessing the level of neurodegenerative changes in selected tissues. The brain and gut are connected by a two-way communication network called the “gut–brain axis”, which is why these tissues have been analyzed for changes in neurometabolism. The significance of the form and dosage of dietary copper and their influence on the functioning of the CNS and peripheral nervous system (PNS) is a widely discussed topic [15,17] in contemporary medical science. According to Rossi et al. [25], copper deficiency impairs the functioning of the mitochondria due to the reduced activity of cytochrome C oxidase. It leads to the overproduction of ROS, disturbance of cellular respiration and neurometabolic impairments. Consequently, neurodegenerative changes and cellular apoptosis occur in the CNS and PNS [25]. Brewer et al. [26] and Greenough et al. [27] state that copper toxicity in the brain is caused by the oxidized form of copper ion Cu^2+^ which is transported into the cells via blood circulation. Fahmy et al. [28] report that the excess of inorganic forms of copper is toxic for cells on the account of stimulating excessive production of ROS. Moreover, the overabundance of copper can inactivate certain cellular enzymes, destroy mitochondrial protein complexes, modify low-density lipoproteins, and damage the DNA [10,29,30]. The results of our research showed that administering a diet enriched with CuNPs in a proportion of 6.5 mg/kg decreased the levels of GAChE and βAP and increased the level of AChE both in the brain and jejunum of rats. At the same time, a beneficial increase in the level of AChE in the brain was also found after the use of CuNPs at a dose of 13 mg. Those changes signify an improvement in neuronal condition and health. According to Zhang et al. [31], βAP induces nerve apoptosis by the damage of the neural mitochondria, where it disrupts ATP production and in turn the entire cellular metabolism. Furthermore, the researchers believe that Cu^2+^ not only accelerates and enhances the neurotoxicity of βAP but is also responsible for blocking the neuronal adenosine receptors [8]. The Cu^2+^ ions enhance the catabolism of adenosine, which presents neuroprotective and neuromodulatory properties. Furthermore, adenosine breakdown results in decreasing the energetic state and lowering the pH inside the cell. Such conditions increase the deposition of βAP in the presence of Cu^2+^. In high concentrations of H^+^, the Cu^2+^ ions act as reducing agents and easily bind with βAP while, simultaneously, hydrogen peroxide is produced, and it is a key factor for neurodegeneration [31]. When the ratio of copper to βAP increases, the level of hydrogen peroxide and the production of hydroxyl radicals increase simultaneously. Then, a change is observed in the morphology of β-amyloid aggregates that transform from fibrous to amorphic ones. This process can lead to damaging the protein–lipid cell structures and the genetic material [29,32]. Earlier results obtained by Cendrowska-Pinkosz et al. [33] suggest that copper cations can accumulate in βAP plaques and increase their production. Concurrently, those cations can easily bind with p-Tau which is responsible for the occurrence of fibril degenerations. Moreover, βAP itself can stimulate the emergence of neurotoxic p-Tau deposits. Kaden et al. [34] and Maynard et al. [12] believe that an optimal level of copper can prevent neurodegeneration because copper inhibits the expression of βAP precursor protein (APP), which is a precursor of βAP. Restricting the expression of APP decreases the levels of βAP, and therefore inhibits the emergence of neurodegenerative changes caused by βAP. However, an excess synthesis of APP lowers the levels of LRP1 which can also give rise to neurodegeneration [33]. According to Kitazawa et al. [35], the excess level of Cu^2+^ stimulates the production of free radicals, decreases the levels of LRP1 and increases the levels of βAP in the brain. What is more, βAP stimulates the reduction of Cu^2+^ to Cu^+^ which leads to the production of toxic hydrogen peroxide (H_2_O_2_), and the effect of copper toxicity is very similar to dementia changes in the brain [4]. CuNPs do not have an electrical charge, which means that they have a greater biochemical activity to stimulate the molecular response in the body and to maintain copper homeostasis outside and inside cells than ionic forms with a positive charge. Research by Gao et al. [36] indicates that copper ion forms and copper nanoparticles have a different effect on the expression of copper transport proteins and the relevant genes encoding these proteins. According to Kang et al. [37], nanocopper can induce the expression of a COX chaperone that prevents the aggregation of unfolded polypeptide chains. It can therefore be presumed that because of the use of nanoparticles, there was an increase in COX expression, which resulted in a decrease in the level of β-amyloid and p-Tau. Gao et al. [36] suggest that nano-Cu in the cell increases the expression of transport proteins CTR1, ATP7A and ATP7B and significantly inhibits the expression of divalent metal transporter 1 (DMT1). According to Schlief et al. [38], oxidative stress caused by a deficiency of antioxidants, i.e., glutathione or metallothionein, stimulates the oligomerization of toxic amyloid, resulting in the formation of atherosclerotic plaques. Studies in mouse models have shown that the depletion of the reduced form of glutathione (GSH) is the cause of amyloid oligomerization and the formation of atherosclerotic plaques [38]. Research by Chang et al. [39] shows that the use of CuNPs enhances antioxidant capacity through the synthesis of low-molecular-weight antioxidants, i.e., glutathione, and inhibits lipid oxidation reactions in the body. The premise of the positive effect of nanocopper on the intracellular homeostasis of this element may be the fact that the use of copper nanoparticles instead of copper carbonate favorably protects proteins and DNA in cells against oxidation and nitration processes [40,41].

The excess amount of copper can accumulate in the βAP and stimulate its production [12]. The copper that is accumulated in the βAP plaques between the neurons significantly restricts interneural communication and accelerates the progression of neurodegeneration. According to Nakamura et al. [42], the complex of Cu^2+^ and βAP produces more ROS compared with free Cu^2+^ ions. A notable outcome of our study is the increase in AChE resulting from administering 6.5 mg of CuNPs in the rat’s diet. In the nerve tissue, AChE deficiency is always connected with neuron dysfunction. What is more, AChE is the main protein of cholinergic synapses in the brain and the neuro-muscular junctions. The lowering of the levels of AChE is considered an early symptom of neuropathology. Underhill et al. [43] demonstrated that an excess concentration of acetylcholine increases protein phosphorylation, including Tau protein. According to Gromadzka et al. [4], excess inorganic copper in the diet favors the deposition of pathologically structured proteins, such as βAP and Tau protein, which gives rise to neurodegenerative changes and leads to cell death. The same authors demonstrated that copper has a strong affinity to tyrosine residues located at the N-terminus of βAP. The oxidative modification of N-terminus tyrosine residues caused by copper induces further oligomerization of βAP [30,32,44,45,46]. This promotes the production of amyloid plaques and Tau protein tangles. Regardless of the applied dosage, the administration of CuNPs in our research caused a very favorable decrease in the levels of Tau protein in rat brains. Furthermore, in our research, we observed that the level of Tau protein was significantly lower in the jejunum of rats who received 6.5 mg of copper compared with animals who received 13 mg of copper in their diet. Therefore, achieving a lower concentration of Tau protein after administering the lower dosage of CuNPs can be considered a successful outcome of the study.

Results of multiple studies showed that the formation of GAChE along with the lowered concentration of AChE leads to notable structural and functional disruptions of the mitochondria, increases the production of ROS, and intensifies apoptotic cell death [21,43,45]. The biochemical changes caused by AChE deficiency and synthesis of GAChE are strongly connected with the disrupted functioning of the N-Methyl-D-Aspartate Receptor (NMDA). The significant decrease in AChE and accumulation of GAChE leads to uncontrolled opening of the calcium channels and calcium influx into the cell. In the mitochondria, it causes excessive activation of CAMK2a. In our research, administration of the recommended dosage (6.5 mg) of copper regardless of its chemical form resulted in obtaining the most beneficial levels of CAMK2a in the brains of rats. CAMK2a is an enzyme responsible for the formation of a complex containing calcium and calmodulin. This can lead to an overaccumulation of Ca^2+^ in the mitochondria and disrupt oxidative phosphorylation and, in turn, induce oxidative stress which gives rise to several pathologies. The low efficiency of oxidative phosphorylation decreases the production of ATP. As a result, it impairs neurometabolism and induces slow nerve cell death [30]. What is more, the complex of calmodulin and calcium synthesized in compliance with CAMK2a during protein kinase activation stimulates pathological protein phosphorylation on the mitochondrial membrane, including Tau protein [42,43,44]. The excessive influx of calcium ions elevates the levels of ROS. As a result, it activates caspase 3 and leads to apoptotic cell death [22,43,44,45].

In our study, we observed low levels of CAMK2a along with decreases in the levels of βAP, Tau protein and GAChE as well as higher levels of AChE in the group of animals who received 6.5 mg of copper in their diet. This leads to the conclusion that the dosage of 6.5 mg of copper is far more effective in preventing neurometabolic impairments compared with 13 mg of copper. Moreover, administering a lower dosage of copper (6.5 mg) led to a favorable increase in the levels of LRP1 in the rat jejunum. Notably, the chemical form of administered copper did not influence the size of this marker. Gao et al. [24] believe that LRP1 captures Tau and p-Tau protein oligomers. Previous research by Cendrowska-Pinkosz et al. [33] showed that administering 6.5 mg of CuNPs per kilogram of diet increases the concentration of LRP1 in the rat tissues. Those results can be further confirmed by Kitazawa et al. [35], who state that the excess of Cu^2+^ stimulates the production of ROS, which is followed by a significant decrease in LRP1 and an increased level of βAP in the brain. In our study, the administration of a recommended dose of copper (6.5 mg) regardless of its chemical form resulted in obtaining the most favorable level of p-Tau. Research conducted by Voss et al. [47] indicates that an excessive amount of copper cations stimulates Tau protein phosphorylation in the brain. However, the authors state that the modulatory mechanism of Tau protein phosphorylation and p-Tau production is yet unclear. On the other hand, it is believed that certain protein kinases are strongly engaged in Tau protein phosphorylation [32,47,48]. According to Lukács et al. [49], the phosphorylation of Tau protein stimulated by the excess of Cu^2+^ occurs in slightly acidic pH during the deprotonation of carboxyl groups of glutamine and aspartyl residues [24]. The results of our study showed that administering 6.5 mg of copper in the diet of the examined rats resulted in a very favorable increase in ABC levels in the brain. In this case, the same effect was achieved regardless of the chemical form of the administered copper. ABC transporters are responsible for the transport of ATP inside the neuron. They are strongly involved in maintaining adequate homeostasis in the brain by eliminating toxic peptides such as βAP, Tau protein or p-Tau which cumulate in the brain along with the development of neurodegenerative changes [50,51,52]. Another favorable result of our study is the lack of copper’s stimulatory influence on DNA methylation. According to Colucci-D’Amato et al. [53] and Nicoletti et al. [54], copper is responsible for modulating the signaling cascades induced by the brain-derived neurotrophic factor (BDNF).

Despite the lack of negative effects of nanocopper on the health condition of rats, a valuable aspect of future research may be the assessment of the pharmacokinetics of nanocopper administration.

## 4. Materials and Methods

### 4.1. Characterization of Cu Nanoparticles

Nanoparticles’ characterization: copper nanoparticles were obtained from Sky Spring Nanomaterials Inc. (Houston, TX, USA), with a purity of 99.9%, 40–60 nm in size (nanopowder), spherical morphology, 0.19 g/cm^3^ bulk density and 8.9 g/cm^3^ true density. A stock solution (5 g/L) was prepared in rapeseed oil, and 9% of the NPs were dissolved as Cu^2+^ ions. The final suspension contained CuNPs and released copper species. The zeta potential of the CuNP suspension was determined to be −30.3 mV (in phosphate buffered saline; PBS) and −38.3 mV (pH 5), and the size was 104 nm (in rapeseed oil), determined by dynamic light scattering with a Zetasizer Nano ZS (Malvern Instruments, Malvern, UK). Cu carbonate (purity ≥ 99%) was sourced from POCH (Gliwice, Poland).

### 4.2. Animal Protocol and Dietary Treatments

The experiment was performed using healthy outbred male Wistar rats (Cmdb:Wi CMDB) fed a semipurified rat diet (Table 5) with two dosages of additional Cu used as CuCO_3_ (CuSALT) and CuNPs (in both cases: standard and two times higher dosages).

The rats aged 6 weeks were randomly divided into 4 groups with n = 10 per 1 group. All animal care and experimental protocols complied with the current laws governing animal experimentation in the Republic of Poland and were approved by an ethical committee according to the European Convention for the Protection of Vertebrate Animals used for Experimental and other Scientific Purposes, Directive 2010/63/EU for animal experiments, and they were approved by the Local Ethics Committee for Animal Experiments. Rats were housed randomly and individually in stainless steel cages under a stable temperature (22 ± 1 °C), relative humidity 60 ± 5%, a 12 h light–dark cycle and a ventilation rate of 15 air changes per hour. All animals throughout the study were monitored daily by the vet for any abnormal behavior to respect the humane endpoints in animal research. The rats were also carefully observed by trained technical staff to recognize any indicators of the animals’ fear, distress, pain, or anxiety. For 9 weeks, the rats had free access to tap water and semipurified diets, which were prepared and then stored at 4 °C in hermetic containers until the end of the experiment. Before the start point, for 5 days, all rats were fed the typical casein-based diet (see Table 5). The diets were modifications of a casein diet for laboratory rodents recommended by the American Institute of Nutrition. In the study, 4 experimental treatments were used to evaluate the effects of two levels of copper carbonate (CuSALT, standard and double dosages 6.5 and 13 mg/kg diet) and two levels of copper nanoparticles (CuNPs, standard and double dosages 6.5 and 13 mg/kg diet), used as an addition to the base feed (see Table 6 and Table 7). All physiological measurements were performed for each animal separately (n = 10 for each group).

### 4.3. Experimental Procedures in Rats and Study Analysis

After the experimental feeding, the rats were anesthetized with a ketamine/xylazine mixture and then they were euthanized by cervical dislocation, according to the accepted EU ethical procedure for laboratory rodents. Then, the brain and jejunum were removed and weighed, from which homogenates were prepared. Homogenates from the brain and jejunum were prepared in accordance with the analytical procedure recommended by the manufacturer of the ELISA enzyme immunoassay, by means of which selected indicators in the tissues were determined. For this purpose, 1 g of brain tissue was weighed and homogenized in ice with 9 mL PBS (0.02 mol/L, pH 7.2) using a porcelain knife. The resulting mixture was then centrifuged for 15 min at 1500× *g*. Next, the supernatant was removed, assayed immediately or the aliquot samples were stored at −80 °C until analysis time.

### 4.4. Determination of Indicators Proving the Potential Neurodegenerative Effect

All analyzed indicators were determined in homogenates of brain and jejunum. Each sample was analyzed in 6 replicates. The research used kits for enzyme immunoassay analysis, by using the ELISA test. An RT6900 microplate reader (DRG MedTek, Warszawa, Poland) was used to read absorbance by the immunoenzymatic method (double binding test—sandwich ELISA). The level of acetylcholinesterase (AChE) was determined by using the Rat Acetylcholinesterase ELISA Kit (Bioassay Technology Laboratory, Inc., Shanghai, China) and the level of amyloid-β using the Rat Total β amyloid Protein (βAP) ELISA Kit produced by Blue Gene Biotech (Shanghai, China). The level of glycosylated acetylcholinesterase (GAChE) was determined using the Rat Glycosylated Acetylcholinesterase ELISA Kit produced by Blue Gene Biotech (Shanghai, China). Tau protein was determined by using the Rat Tau Protein (Tau) ELISA Kit, and the level of phosphorylated p-Tau protein was determined using the Rat Phosphorylated tau 231 (p-Tau 231) ELISA (Blue Gene Biotech, Shanghai, China). Low-density lipoprotein receptor-related protein 1 (LRP1) was determined using a Rat Low-Density Lipoprotein Receptor Related Protein 1 ELISA Kit produced by Blue Gene Biotech (Shanghai, China). ATP-binding cassette transporters were determined by Rat ATP-Binding Cassette Transporter G2 ELISA Kit, and kinase II was determined using the Rat Calcium/Calmodulin-Dependent Protein Kinase II Alpha (CAMK2a) ELISA Kit (Blue Gene Biotech, Shanghai, China). The levels of epigenetic changes in the brain and jejunum were determined by analyzing global DNA methylation (methylome) using diagnostic kits from Sigma Aldrich, St. Louis, MO, USA.

### 4.5. Statistical Analysis

Prior to ANOVA analyses, the normality and homogeneity of variance were verified by the Shapiro–Wilk and the Levene tests, respectively. The data were calculated statistically with the aid of two-way ANOVA, where the main analyzed factors were the additional dietary copper source (S; CuSALT—copper carbonate (CuCO_3_); CuNPs—copper nanoparticles) and the dosage of copper addition (D; 6.5 and 13 mg/kg of a diet). The interaction of two-way ANOVA S×D was also calculated. In the case of a significant interaction effect, the Newman–Keuls test was used to evaluate the differences between the four experimental groups. The GLM procedure in Statistica 13.0 PL software ( StatSoft Corp^®^, Krakow, Poland) was used for the statistical analysis. The differences were considered significant at *p* ≤ 0.05. All data were expressed as mean values with pooled standard error (SEM).

## 5. Conclusions

The results of this study confirmed that neuroprotective properties of copper can be enhanced if the traditional carbonate form (CuCO_3_) is exchanged for CuNPs. However, it is not necessary to increase the recommended dosage from 6.5 mg to 13 mg because 6.5 mg has a more beneficial impact on restricting the development of neurodegenerative changes. The neuroprotective effect of CuNPs in dosage of 6.5 mg is proven by the increased levels of AChE and decreased levels of GAChE and βAP in the brain as well as lowered levels of Tau protein and increased levels of LRP1 in the jejunum. The studies showed no negative effect of the dose or form of copper on the health condition of the rats, and no decrease in the final body weight of the rats was noted. The animals also did not show any negative behavioral features associated with receiving copper nanoparticles. Without a doubt, additional research is required to understand how dietary nanoparticles operate in humans, and it is advised to use caution while informing the public about a new and promising source of dietary nanoparticles [55].

## Figures and Tables

**Table 1 ijms-24-11321-t001:** Levels of selected indices of neural tissue metabolism in brain tissue, part 1.

Treatment	AChE (ng/mL)	GAChE (ng/mL)	LRP1 (pg/mL)	ABC (ng/mL)	% Methylation
CuSALT_6.5_	12.59 ± 0.451	2.31 ± 0.017	8.35 ± 0.078	2.65 ± 0.024	75.58 ± 0.474
CuSALT_13_	9.94 ± 0.365	2.46 ± 0.069	7.51 ± 0.092	1.26 ± 0.034	79.36 ± 0.431
CuNPs_6.5_	12.16 ± 0.541	0.68 ± 0.028	11.27 ± 0.074	2.79 ± 0.079	75.11 ± 0.245
CuNPs_13_	14.58 ± 0.325	1.87 ± 0.046	7.65 ± 0.065	1.63 ± 0.062	79.21 ± 0.247
Effect source (S)					
CuSALT	11.27	2.39 ^a^	7.93	1.96	77.47
CuNPs	13.37	1.28 ^b^	9.46	2.21	77.16
Effect dose (D)					
6.5 mg/kg	12.34	1.49 ^b^	9.81	2.72 ^a^	75.35
13 mg/kg	12.26	2.17 ^a^	7.58	1.45 ^b^	79.29
*p*-value					
S effect	0.058	0.009	0.082	0.462	0.196
D effect	0.165	0.026	0.075	0.033	0.078
S × D	0.012	0.623	0.792	0.126	0.427

^a,b^—mean values ± SEM within a column with unalike superscript letters differ significantly (*p* < 0.05). CuSALT—copper carbonate (CuCO_3_); CuNPs—copper nanoparticles. CuSALT—a group receiving a diet resulting in 6.5 or 13 mg copper from CuCO_3_ per 1 kg of diet; CuNPs_6.5_—a group receiving a diet containing 6.5 mg copper from the Cu nanoparticles’ preparation per 1 kg of diet, CuNPs_13_—a group receiving a diet containing 13 mg copper from the Cu nanoparticles’ preparation per 1 kg of diet, AChE—acetylcholinesterase; GAChE—glycosylated acetylcholinesterase; LRP1—low-density lipoprotein receptor-related protein 1; ABC—ATP-binding cassette transporter.

**Table 2 ijms-24-11321-t002:** Levels of selected indices of neural tissue metabolism in jejunum tissue, part 1.

Treatment	AChE (ng/mL)	GAChE (ng/mL)	LRP1 (pg/mL)	ABC (ng/mL)	% Methylation
CuSALT_6.5_	10.49 ± 0.147	3.72 ± 0.453	2.86 ± 0.418	0.56 ± 0.017	74.37 ± 0.721
CuSALT_13_	6.39 ± 0.231	4.43 ± 0.789	2.78 ± 0.731	1.26 ± 0.013	79.36 ± 0.624
CuNPs_6.5_	16.36 ± 0.245	0.68 ± 0.654	5.34 ± 0.695	0.34 ± 0.019	72.57 ± 0.614
CuNPs_13_	13.41 ± 0.214	1.87 ± 0.843	3.35 ± 0.413	1.04 ± 0.013	74,11 ± 0.781
Effect source (S)					
CuSALT	8.44 ^b^	3.85 ^a^	2.82 ^b^	0.91	76.87
CuNPs	14.89 ^a^	1.28 ^b^	4.35 ^a^	0.69	73.34
Effect dose (D)					
6.5 mg/kg	13.43 ^a^	2.19 ^b^	4.01 ^a^	0.45 ^b^	73.47
13 mg/kg	9.89 ^b^	3.15 ^a^	3.01 ^b^	1.15 ^a^	76.74
*p*-value					
S effect	0.024	0.007	0.043	0.073	0.137
D effect	0.006	0.021	0.002	0.001	0.095
S × D	0.247	0.092	0.088	0.456	0.093

^a,b^—mean values ± SEM within a column with unalike superscript letters differ significantly (*p* < 0.05). CuSALT—copper carbonate (CuCO_3_); CuNPs—copper nanoparticles. CuSALT—a group receiving a diet resulting in 6.5 or 13 mg copper from CuCO_3_ per 1 kg of diet; CuNPs_6.5_—a group receiving a diet containing 6.5 mg copper from the Cu nanoparticles’ preparation per 1 kg of diet, CuNPs_13_—a group receiving a diet containing 13 mg copper from the Cu nanoparticles’ preparation per 1 kg of diet, AChE—acetylcholinesterase; GAChE—glycosylated acetylcholinesterase; LRP1—low-density lipoprotein receptor-related protein 1; ABC—ATP-binding cassette transporter.

**Table 3 ijms-24-11321-t003:** Levels of selected indices of neural tissue metabolism in brain tissue, part 2.

Treatment	βAP pg/mL	Tau ng/mL	p-Tau ng/mL	CAMK2a ng/mL
CuSALT_6.5_	10.49 ± 0.258	1.36 ± 0.254	0.24 ± 0.087	0.93 ± 0.047
CuSALT_13_	15.59 ± 0.457	2.29 ± 0.147	0.38 ± 0.047	1.34 ± 0.046
CuNPs_6.5_	9.75 ± 0.089	0.98 ± 0.095	0.15 ± 0.036	0.40 ± 0.015
CuNPs_13_	12.76 ± 0.584	1.25 ± 0.087	0.33 ± 0.047	1.22 ± 0.078
Effect source (S)				
CuSALT	13.04 ^a^	1.83 ^a^	0.43	0.67
CuNPs	11.07 ^b^	1.16 ^b^	0.24	0.81
Effect dose (D)				
6.5 mg/kg	10.12 ^b^	1.17	0.19 ^b^	0.67 ^b^
13 mg/kg	14.18 ^a^	1.77	0.34 ^a^	1.28 ^a^
*p*-value				
S effect	0.017	0.001	0.065	0.097
D effect	0.036	0.072	0.003	0.005
S × D	0.135	0.063	0.182	0.236

^a,b^—mean values ± SEM within a column with unalike superscript letters differ significantly (*p* < 0.05). CuSALT—copper carbonate (CuCO_3_); CuNPs—copper nanoparticles. CuSALT—a group receiving a diet resulting in 6.5 or 13 mg copper from CuCO_3_ per 1 kg of diet; CuNPs_6.5_—a group receiving a diet containing 6.5 mg copper from the Cu nanoparticles’ preparation per 1 kg of diet, CuNPs_13_—a group receiving a diet containing 13 mg copper from the Cu nanoparticles’ preparation per 1 kg of diet, βAP—β amyloid protein; Tau—Tau protein; p-Tau—hyperphosphorylated Tau protein; CAMK2a—Calcium/Calmodulin-Dependent Protein Kinase II Alpha.

**Table 4 ijms-24-11321-t004:** Levels of selected indices of neural tissue metabolism in jejunum tissue, part 2.

Treatment	βAP (pg/mL)	Tau (ng/mL)	p-Tau (ng/mL)	CAMK2a (ng/mL)
CuSALT_6.5_	10.19 ± 0.124	1.67 ± 0.458	0.642 ± 0.015	0.097 ± 0.061
CuSALT_13_	13.28 ± 0.213	1.38 ± 0.325	1.067 ± 0.078	0.144 ± 0.041
CuNPs_6.5_	7.45 ± 0.951	0.81 ± 0.475	0.054 ± 0.036	0.097 ± 0.021
CuNPs_13_	10.46 ± 0.478	2.77 ± 0.461	0.885 ± 0.048	0.098 ± 0.027
Effect source (S)				
CuSALT	11.74 ^a^	2.83	0.86	0.12
CuNPs	9.11 ^b^	1.79	0.72	0.09
Effect dose (D)				
6.5 mg/kg	8.82 ^b^	1.24 ^b^	0.35	0.09
13 mg/kg	11.87 ^a^	3.38 ^a^	0.98	0.16
*p*-value				
S effect	0.005	0.057	0.108	0.082
D effect	0.007	0.049	0.062	0.078
S × D	0.378	0.742	0.207	0.063

^a,b^—mean values ± SEM within a column with unalike superscript letters differ significantly (*p* < 0.05). CuSALT—copper carbonate (CuCO_3_); CuNPs—copper nanoparticles. CuSALT—a group receiving a diet resulting in 6.5 or 13 mg copper from CuCO_3_ per 1 kg of diet; CuNPs_6.5_—a group receiving a diet containing 6.5 mg copper from the Cu nanoparticles’ preparation per 1 kg of diet, CuNPs_13_—a group receiving a diet containing 13 mg copper from the Cu nanoparticles’ preparation per 1 kg of diet, βAP—β amyloid protein; Tau—Tau protein; p-Tau—hyperphosphorylated Tau protein; CAMK2a—Calcium/Calmodulin-Dependent Protein Kinase II Alpha.

**Table 5 ijms-24-11321-t005:** Composition of basal diet fed to rats, %.

Basal Diet	%
Ingredient	
Casein ^a^	14.8
DL-methionine	0.2
Cellulose ^b^	8.0
Choline chloride	0.2
Rapeseed oil	8.0
Cholesterol	0.3
Vitamin mix. ^c^	1.0
Mineral mix. ^d^	3.5
Maize starch ^e^	64.0

^a^ Casein preparation: crude protein 89.7%, crude fat 0.3%, ash 2.0% and water 8.0%. ^b^ α-Cellulose (SIGMA, Poznan, Poland), main source of dietary fiber. ^c^ AIN-93G-VM [24], g/kg mix: 3.0 nicotinic acid, 1.6 Ca pantothenate, 0.7 pyridoxine-HCl, 0.6 thiamin-HCl, 0.6 riboflavin, 0.2 folic acid, 0.02 biotin, 2.5 vitamin B-12 (cyanocobalamin, 0.1% in mannitol), 15.0 vitamin E (all-rac-α-tocopheryl acetate, 500 IU/g), 0.8 vitamin A (all-trans-retinyl palmitate, 500,000 IU/g), 0.25 vitamin D3 (cholecalciferol, 400,000 IU/g), 0.075 vitamin K-1 (phylloquinone), 974.655 powdered sucrose. ^d^ Variable component in relation to copper level, mineral mixture (base according to NRC [24]) with different Cu levels from two sources (standard source CuCO_3_ and experimental preparation of Cu nanoparticles), see Table 6 and Table 7. ^e^ Maize starch preparation: crude protein 0.6%, crude fat 0.9%, ash 0.2%, total dietary fiber 0%, water 8.8%.

**Table 6 ijms-24-11321-t006:** The experimental scheme of applied copper ^a^ (provided Cu dosage was calculated considering CuCO_3_ in mineral mixture or copper from CuNPs’ preparation).

Treatment	Source of Cu
CuSALT_6.5_	a diet containing 6.5 mg/kg Cu from CuCO_3_ (n = 10)
CuSALT_13_	a diet containing 13 mg/kg Cu from CuCO_3_ (n = 10)
CuNPs_6.5_	a diet containing 6.5 mg/kg Cu from Cu nanoparticles’ preparation (n = 10)
CuNPs_13_	a diet containing 13 mg/kg Cu from Cu nanoparticles’ preparation (n = 10)

n = 10, number of rats used feeding period. ^a^ Experimental groups: CuSALT_6.5_—the rats were fed a diet with standard mineral mixture resulting in 6.5 mg Cu (from CuCO_3_ in mineral mixture) per 1 kg of diet during 2 months of feeding; CuSALT_13_—the rats were fed a diet with double dosages of mineral mixture resulting in 13 mg Cu (from CuCO_3_ in mineral mixture) per 1 kg of diet during 2 months of feeding; CuNPs_6.5_ and CuNPs_13_—the rats were given a diet containing 6.5 or 13 mg/kg Cu from Cu nanoparticles’ preparation per 1 kg of diet during 2 months of feeding; CuNPs_13_ the rats were given a diet containing 13 mg/kg Cu from Cu nanoparticles’ preparation per 1 kg of diet during 2 months of feeding.

**Table 7 ijms-24-11321-t007:** Composition of mineral mixtures used in experimental diets, g/kg.

	Mineral Mixture with CuSALT	Mineral Mixture Deprived of CuSALT
Calcium carbonate anhydrous CaCO_3_	357	357
Potassium phosphate monobasic K_2_HPO_4_	196	196
Potassium citrate C_6_H_5_K_3_O_7_	70.78	70.78
Sodium chloride NaCl	74	74
Potassium sulfate K_2_SO_4_	46.6	46.6
Magnesium oxide MgO	24	24
Microelements mixture	18	18
Starch	213.62	213.62
Microelements mixture, g/100 g		
Ferric citrate (16.7% Fe)	31	31
Zinc carbonate ZnCO_3_ (56%Zn)	4.5	4.5
Manganous carbonate MnCO_3_ (44.4% Mn)	23.4	23.4
Copper carbonate CuCO_3_ (55.5% Cu)	1.85 ^a^/3.7 ^b^	0 ^c^
Potassium iodate KJ	0.04	0.04
Citric acid C_6_H_8_O_7_	40.7 g	40.7 g

^a^ given to CuSALT_6.5_ groups (2 months of feeding); diet containing 6.5 mg/kg Cu from CuCO_3_. ^b^ given to CuSALT_13_ groups (2 months of feeding); diet containing 13 mg/kg Cu from CuCO_3_. ^c^ given to CuNPs groups; during 2 months of feeding, the groups CuNPs_6.5_ and CuNPs_13_ were provided with an appropriate amount of Cu from Cu nanoparticles’ preparation as an emulsion along with dietary rapeseed oil.

## Data Availability

Data supporting reported results are available on request.

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
