# Peer review of "The Influence of Copper Nanoparticles on Neurometabolism Marker Levels in the Brain and Intestine in a Rat Model"

_ijms, 2023, doi:10.3390/ijms241411321_

Round 1

Reviewer 1 Report

The manuscript deals with the effect of different copper species (CuCO3 or CuNPs) and doses on protective and toxic markers describing the neurodegrnerative status in the brain and jejunum. The findings indicate that the neuroprotective effect of copper can be enhanced by replacing the carbonate form with nanoparticles and utilizing the lower dose of employed copper. The analytical results appear reliable and the invoked biomarkers appropriates. My concern regards the significance of the use of the periferal copper form to attribute the species to employ. This can be considered a sort of tip of icebergs due to the fact that the external metal stimuli affect the intracellular metal level where the copper is presents mainly as Cu+ and is strongly regulated by a network of metal transporters, chaperones, storages and small molecules as glutathione, etc. The relationship between extracellular and intracellular copper homeostasis should be considered to define in more significant mode the species involved in the levels of biomarkers reported in the study(about the metal homeostasis, see Coordination Chemistry Reviews 433 (2021) 213727. doi:10.1016/j.ccr.2020.213727 or Nat Rev Cancer 22 (2022) 102. Doi: 10.1038/s41568-021-00417-2 and references therein). My suggestion is to revise the study focusing on the players of intracellular copper homeostasis responsible of the biomarkers expression

Author Response

Dear Reviewer
We thank you very much for the substantive review. We hope that we have responded satisfactorily to the comments provided by the Reviewer. At the same time, we thank you for your valuable tips and advice.

Reviewer 2 Report

The article by Cendrowska-Pinkosz present interesting results on dietary supplementation of Cu2+ to improve neuroprotection. The introduction and the results are well presentend. Additional data on mice health follow up (i.e. weight) can be included in the article (if available) to evaluate possible toxicity. 

I would also add a comment in the discussion section on the necessity to study the pharmacokinetic of copper supplements administration to evaluate if there could be long term toxic effects for copper accumulation. 

Author Response

(The authors gave the same response as above.)

Reviewer 3 Report

·   Dear author and co-authors:

Here are my comments to be considered:  

  I I suggest to shorten the title of the publication.

·       Line 14: use present form: is instead of was.

·       Line 47: add references for the ‘current reports’

·       General comment: try to write shorter sentences. For example: paragraph 70-86 is too long and you use long sentences.

·       Consider rephrasing some sentences to enhance readability and flow. For example, instead of "Due to the potential neurotoxic effects of a diet with excessive copper," you could write "Because excessive copper in the diet can have neurotoxic effects." See discussion section

·       Consider expanding the conclusions or merge it with discussion section.

Minor editing of English language required

Author Response

(The authors gave the same response as above.)

Reviewer 4 Report

Dear the EditorCendrowska-Pinkosz, M et al reported the effect of different formulation of copper in rat diet for proteinous parameters in the brain and jejunum tissues. In the result section, the manuscript appeared to be ill-prepared without providing sufficient intoroduction of why each experiment was performed. Text was too short not to fully describe the experimental outcome of Tables 1-4.Major concerns:1) Text should be rewrote by an English-native writer.2) Please write Discussion more concisely by only citing essential references.3) In Tables, each measurement should have own SEM. In this study, SEM for a group of differennt treatments was calculated.

Text should be rewrote by an English-native writer.

Author Response

(The authors gave the same response as above.)

Round 2

Reviewer 1 Report

The need to find the right biomarker among the different players of copper homeostasis is clarified also by the cited references reported in the letter of the authors. The statement to presume or to assume is an honest answer but it is not acceptable. The reply of the authors to my comments summarized in the following clearly indicates the reasons for which the manuscript cannot be recommended for publication.

The reviewer recommends revising the study to determine the expression of the analyzed biomarkers. We are currently unable to do this, but we will keep this suggestion in mind in our future research. Despite the impossibility of assessing gene expression, we can presume that nanocopper influenced the expression of proteins responsible for intracellular copper homeostasis.

The results of our research allow us to assume that the observed beneficial changes in the level of markers indicating the decrease of neurometabolism disorders may be the result of improved copper homeostasis outside and inside cells.

Author Response

Dear Reviewer

Thank you for your review. We respect the reviewer's opinion. We understand the need to enrich research with molecular analyses.   When establishing the research hypothesis, we had our own vision of research, which we implemented. We cannot supplement the research proposed by the Reviewer, among others due to the lack of biological material. However, we will keep the suggested research in mind in the future. Thank you for taking the time to review our manuscript.

Reviewer 4 Report

Dear the Editor,

This revised manuscript extentively answered all raised concerns by the Reviewer.

Author Response

Dear Reviewer

We are glad this revised manuscript comprehensively answered all the doubts raised by the Reviewer. Thank you for a substantive and transparent review.
